# The Crossroads between Infection and Bone Loss

**DOI:** 10.3390/microorganisms8111765

**Published:** 2020-11-10

**Authors:** Tiago Carvalho Oliveira, Maria Salomé Gomes, Ana Cordeiro Gomes

**Affiliations:** 1i3S—Instituto de Investigação e Inovação em Saúde, Universidade do Porto, 4200-135 Porto, Portugal; tcoliveira@ibmc.up.pt (T.C.O.); sgomes@ibmc.up.pt (M.S.G.); 2Faculdade de Ciências da Universidade do Porto, 4169-007 Porto, Portugal; 3Instituto de Ciências Biomédicas de Abel Salazar da Universidade do Porto, 4050-313 Porto, Portugal

**Keywords:** bone, infection, chronic inflammation, mesenchymal stem cell, osteoclasts, osteoblasts

## Abstract

Bone homeostasis, based on a tight balance between bone formation and bone degradation, is affected by infection. On one hand, some invading pathogens are capable of directly colonizing the bone, leading to its destruction. On the other hand, immune mediators produced in response to infection may dysregulate the deposition of mineral matrix by osteoblasts and/or the resorption of bone by osteoclasts. Therefore, bone loss pathologies may develop in response to infection, and their detection and treatment are challenging. Possible biomarkers of impaired bone metabolism during chronic infection need to be identified to improve the diagnosis and management of infection-associated osteopenia. Further understanding of the impact of infections on bone metabolism is imperative for the early detection, prevention, and/or reversion of bone loss. Here, we review the mechanisms responsible for bone loss as a direct and/or indirect consequence of infection.

## 1. Introduction

Invading pathogens such as bacteria, viruses, and parasites are capable of colonizing several tissues within the human body, namely the bone, eliciting an immune response in the majority of immune competent individuals. Indeed, there is some evidence that viruses such as hepatitis B and C viruses and human immunodeficiency virus disrupt bone metabolism causing bone loss [1,2,3,4,5,6]. On the other hand, bacteria such as *Borrelia burgdorferi*, *Mycobacterium tuberculosis,* and *Staphylococcus aureus* are causative agents of osteomyelitis and subsequently lead to bone destruction [7,8,9]. While some pathogens may colonize the bone cells triggering bone degradation, the production of pro-inflammatory mediators such as cytokines and growth factors may also be responsible for the dysregulation of bone homeostasis, namely in situations such as sepsis [10]. Hence, bone loss in response to infection may occur either directly due to the colonization of bone by microorganisms or indirectly due to immunopathology. Here, we address the cellular and molecular mechanisms responsible for reduced bone mass during infection.

Bone immunopathology is related to the production of immune mediators that are sensed by bone cells, altering their physiology. Along these lines, it is conceivable that the aggressions to bone homeostasis induced by infection have either to reach a high magnitude or to be kept constant throughout a long period of time so as to induce a detectable phenotype or clinical manifestation. These characteristics are more similar with the alterations observed during chronic inflammatory diseases. Chronic inflammatory diseases enclose a wide range of disorders in the immune system caused by the uncontrolled immune response to the self (autoimmunity); a persistent or difficult to eradicate pathogen; or even by a mix of the former situations. While some of the mechanisms involved in autoimmunity and bone pathologies have been well studied/characterized in pathologies such as rheumatoid arthritis [11], the mechanisms responsible for bone pathology induced by chronic bacterial infection have not been analyzed. Yet, there is a substantial overlap between the immune mediators produced during autoimmunity and chronic infection. In the present review, we also focus on the role of immune mediators produced in response to bacterial infection that may impact the physiology of bone-related cellular components. Therefore, we aim to identify the known direct and indirect cellular and molecular pathways involved in the interactions between microorganisms and bone cells. This will be interesting for basic sciences researchers who study several aspects of infection and immunopathology as their in vivo infection models could also be used to highlight other alterations in bone metabolism during infection. The clinical management of bone infections such as antibiotics regimens and/or surgical debridement of the infected bones and prothesis implantations are out of the scope of the present review. Nevertheless, before addressing the impact of infection in bone metabolism, it is key to understand the pathways of bone formation and degradation as well as the mechanisms regulating these pathways, which are covered in the second section of the present manuscript. Then, the mechanisms responsible for bone loss during infection are discussed in Section 3 and Section 4.

## 2. Bone Structure and Metabolism

The bone is a type of specialized, strong, and rigid connective tissue and is a major component of the musculoskeletal system. These characteristics allow the terrestrial locomotion by providing support and protection to the body. Yet, it is a very dynamic tissue, being highly vascularized and populated with high numbers of cells compared to other tissues in the body. Together, these features enable its adaptation to changing mechanical loads and its regeneration following injury.

Anatomically, the bone is usually described at the macroscopic and the microscopic levels. Macroscopically, most of the long bones of mammals are divided into compact and trabecular bones (Figure 1a). The former is dense, while the latter is spongy and honeycombed by large bone marrow cavities, where the hematopoietic and fat tissues are stored. The boundary between the hematopoietic tissues within the bone cavities and the bone itself is denominated endosteum (Figure 1b), which contains bone forming cells and their progenitors and is important for calcium homeostasis. On the other hand, lining the external bone surface (except in articular surfaces and regions of attachments of tendons and ligaments) is the periosteum (Figure 1b). This zone is rich in collagen fibers and important for appositional growth during fetal development and for fracture healing during adult life. While the endosteum lines both the trabecular and compact bones, the periosteum only exists in the outer surface of compact bone. Besides their different aspects, compact and trabecular bones have different functions and localizations: the compact bone is present mostly in the diaphysis, providing thickness and strength in bending, whereas in the epiphysis, abundant trabecular bones provide strength in compression (Figure 1a). Furthermore, the two types of bone play distinct roles during growth: compact bone is responsible for appositional growth and trabecular bone for the longitudinal growth [12].

Zooming in on the microstructure of the bone, this specialized connective tissue is composed by specialized cells meshed within an extracellular matrix rich in water, type I collagen, and hydroxyapatite. The cellular components of the bone may be grouped into two interconnected arms: the cells responsible for bone deposition—the osteoblasts—and the cells capable of degrading the mineral matrix—the osteoclasts (Figure 1b) [12]. Besides the two main cell types, a special subtype of macrophages expressing CD169, the osteomacs, reside nearby the bone surfaces, supporting osteoblast maturation and function under homeostatic or regenerative conditions [13,14,15,16,17,18].

An adequately balanced bone mass is maintained by a tight regulation and crosstalk between the cells forming and degrading the bone. Therefore, any alteration in the physiology of one cell type, besides altering their own physiology, will dysregulate the action of the other type. In this section, we discuss the main types of bone cells and their cross-regulation.

### 2.1. Bone Forming Cells: Osteoblasts and Their Progenitors

Osteoblasts are bone-forming cells layering the surfaces of growing or remodeling bone, in close contact with other bone cells (Figure 1b). These cells are derived from mesenchymal stem cells (MSC) residing in bone marrow and other connective tissues, and their main function is to synthetize, secrete, and deposit the extracellular matrix of the bone. MSC have the multipotency to generate osteoblasts (bone forming cells), adipocytes (fat cells), and chondrocytes (cartilage cells), meanwhile guiding the maintenance and differentiation of hematopoietic stem and progenitor cells [19,20,21,22]. The commitment of MSC to the osteoblastic lineage is thought to be a stepwise process generating progenitors with an increasing degree of lineage commitment and less potency to generate the other lineages [23,24]. These alterations in potency are translated into the differential expression of transcription factors and gene networks as well as the expression of surface markers [25,26,27], namely the upregulation of the transcription factors Runx2 and Osterix. These transcription factors control the expression of bone-matrix-related genes such as collagen type I alpha 1, sialoprotein, osteopontin, and fibronectin [28]. Additionally, the crosstalk between MSC-derived mature cells seems to regulate the MSC differentiation and lineage choice, in order to maintain an adequate balance of bone and fat mass [29]. Of note, recent evidence from in vitro studies suggests that the level of inflammation in the bone marrow environment may also influence the differentiation potential of MSC [30], which could affect the ability of these cells to regenerate bone.

Osteoblasts produce collagens fibers, osteocalcin and osteonectin, which are important players in bone formation. Osteocalcin binds hydroxyapatite and calcium, being fundamental for bone mineralization and a marker for new bone formation. Osteonectin also binds to hydroxyapatite but also to collagen fibers. Osteoblasts also express the ligand for the receptor activator of nuclear factor kappa-B (RANKL), which binds to the receptor activator of nuclear factor kappa-B (RANK) expressed on the surface of osteoclasts and their progenitors or to osteoprotegerin (OPG), a decoy receptor (see below) (Figure 2). In order to produce new bone, osteoblasts secrete osteocalcin and release alkaline phosphatase and pyrophosphatase, overall increasing the local concentration of calcium and phosphate in the newly forming osteoid, initiating the formation of mineral crystals. Once the newly forming osteoid becomes mineralized, osteoblasts become entrapped within the bone matrix, becoming osteocytes. Osteocytes are mature bone cells and lack the capacity to deposit new bone matrix but have several cellular projections that contact with the other bone cells, regulating their physiology (Figure 1b).

### 2.2. Bone Degrading Cells: Osteoclasts

The destruction of bone is accomplished by osteoclasts that are large, multinucleated cells with hematopoietic origin. The classical pathway of osteoclast development in adult bone marrow consist in the differentiation of hematopoietic stem cells (HSC) into myeloid restricted progenitors, the common myeloid progenitors (CMP) that in turn originate monocytes and macrophages. The latter, when stimulated with macrophage-colony stimulating factor (M-CSF) and RANKL form osteoclast progenitors (OCP), which upon further stimulation with RANKL and the occurrence of fusion events originate osteoclasts. Osteoclasts are characterized by the expression of tartrate-resistant acid phosphatase (TRAP), cathepsin K, calcitonin receptor, and α_v_β_3_-integrin [28]. A recent report has challenged this view by demonstrating the embryonical origin of osteoclasts from erythro-myeloid progenitors and the postnatal maintenance of these long-lived cells by the iterative fusion of circulating monocytes with pre-existing osteoclasts [31].

In order to dissolve the mineral matrix, osteoclasts contain several mitochondria and acid phosphatases-rich lysosomes. The latter are transported along a vast microtubule array to the ruffle border of the cell that faces the resorption bay, followed by the secretion of protons by v-ATPases that locally acidifies the environment, causing the resorption of bone minerals such as calcium carbonates and calcium phosphates [32]. This process is coupled with the secretion of cathepsin K and collagenase leading to the dissolution of the organic component of the bone [12].

### 2.3. Crosstalk between Bone Formation and Degradation

Osteoclasts and osteoblasts have opposing functions: the first degrade the bone and the latter form new bone. Nevertheless, both cell types work together in a concerted manner. A delicate and highly regulated interplay between the functions of these two cell types is required in order to maintain a balanced bone mass, thus, preventing the development of osteoporosis (bone mass deficit) or osteopetrosis (excess of bone mass). The crosstalk between bone formation and degradation is summarized in Figure 2.

Under homeostatic conditions, osteoblasts secrete M-CSF, which binds to its receptor on the surface of myeloid cells and osteoclast progenitors. M-CSF promotes the proliferation and survival of osteoclast progenitors [33]. Simultaneously, osteoblasts secrete RANKL that may either bind to the RANK receptor at the surface of osteoclast progenitors or be sequestered by the decoy receptor OPG. Osteoblasts also secrete OPG, whose binding to RANKL, sequesters the latter, inhibiting osteoclastogenesis [34,35]. Together M-CSF and RANKL promote the survival and differentiation of osteoclast progenitors and consequently bone resorption [36]. Therefore, both the enhancement and the inhibition of the differentiation of osteoclasts and their progenitors are controlled by osteoblasts [37].

Not only osteoblasts control osteoclastogenesis. The terminally differentiated bone cells, the osteocytes, are also involved in the crosstalk between bone forming and bone degrading cells by sensing the mechanical load and secreting factors like prostaglandins, ATP, RANKL and Wnt1 proteins that modulate osteoblastogenesis and osteoclastogenesis. The Wnt pathway is fundamental for bone density by acting on osteoblasts and osteocytes [38,39]. Overall, the tight regulation between the different cell types allows bone remodeling.

The process of bone remodeling is fundamental for the healing and response to stress alterations in the bone mass [40]. Besides the soluble mediators discussed before, it is controlled by cell-cell and cell-extracellular matrix interactions [41,42].

The bone is responsible for the storage of calcium and phosphates that are fundamental for several biological processes. Dysregulation in bone homeostasis will affect the metabolism of these minerals and, thus, it is not surprising that bone homeostasis is also regulated by hormones such as the parathyroid hormone (PTH), calcitonin, and mediators such as vitamin D that control the balance of calcium and phosphate in the body.

PTH is secreted in response to systemic hypocalcemia (reduced circulating calcium levels) and according to the mode of secretion may differentially affect bone homeostasis. Persistent PTH secretion drives osteolysis followed by bone resorption. On the other hand, intermittent secretion of PTH initially leads to osteolysis and increased activity of osteoclasts, which in turn will stimulate osteoblast proliferation leading to new bone formation [43,44].

Vitamin D may also directly influence bone metabolism as bone-related cells, namely osteoblasts and their progenitors, express the vitamin D receptor. Indeed, MSC differentiate towards the osteoblast lineage upon stimulation with vitamin D. Furthermore, the synthesis and secretion of osteocalcin, osteopontin, and RANKL by osteoblasts are also regulated by the aforementioned vitamin [43].

Finally, calcitonin, a hormone secreted under hypercalcemia, transiently opposes the actions of PTH in osteoclasts and directly inhibits osteoclast function, thus promoting bone formation. The direct effect of calcitonin on osteoclasts is mediated by the signaling through the calcitonin receptor on the cell surface, which blocks the formation of membrane invaginations and active proton secretion, leading the cell into a quiescent state. Furthermore, calcitonin also arrests osteoclast differentiation at immature stages [43].

Amongst the hormones controlling bone homeostasis are estrogens. Estrogens induce the expression of the Fas ligand (FasL) in osteoclasts and concomitantly apoptosis, thus, decreasing osteoclasts numbers and activity. Thus, deficiency in estrogen either by genetic manipulation of laboratory animal models or in post-menopausal or ovariectomized women correlates with decreased bone mass [45,46]. On the other hand, endogenous glucocorticoids increase the production of RANKL and decrease that of OPG. In this way, they increase the levels of RANKL available for osteoclasts, thus promoting bone resorption [47].

Several other factors control bone metabolism. On the bone formation promoting side, fibroblast growth factors (FGF) 2 and 18 promote MSC differentiation into osteoblasts [48]. Bone morphogenetic proteins (BMP) are released by osteoclasts and act on osteoblasts and chondrocytes as bone deposition promotors [49].

## 3. Bacterial Osteomyelitis

Osteomyelitis is a difficult to treat bone infection in which an invading microorganism colonizes one or several regions of the bone (marrow, cortex, periosteum). This colonization causes a local inflammatory response that leads to the progressive destruction of the bone (Figure 3) [50,51]. Usually, these infections occur following bone trauma or when a high pathogen burden or foreign bodies are present in the body [52,53]. The microorganisms reach the bone through different routes: (1) hematogenous spread usually to the vertebrae in adults or long bones in children; (2) dissemination from a contiguous site after surgery or bone trauma, commonly to the sternum; (3) secondary infection, due to concomitant vascular insufficiency or neuropathy, namely foot infections in diabetic patients.

Several classifications can be applied to osteomyelitis. One of them uses the anatomical site of infection, dividing the infection into vertebral osteomyelitis, long bones osteomyelitis, sternal osteomyelitis, foot osteomyelitis, and periprosthetic joint infection. These different types of osteomyelitis differ in the routes of dissemination used by the pathogen to colonize the bone, the age of the patients, and their associated co-morbidities. For instance, foot osteomyelitis is common in diabetic individuals. Vertebral osteomyelitis, due to surgical interventions, is increasingly important, as the number of these infections acquired in the hospital setting is increasing. The hematogenous seeding of long bones in children when ineffectively treated may progress to chronic osteomyelitis, which can recur after an interval of more than 70 years without symptoms [54].

The duration of the infection is also used to classify osteomyelitis: acute osteomyelitis refers to an infection occurring for several days to weeks, whereas an infection progressing for months to years is categorized as chronic osteomyelitis. The latter usually is associated with the persistence of microorganisms and low-grade inflammation, leading to portions of dead bone (sequestrum) and fistulous tracts. On the other hand, acute osteomyelitis is associated with suppurative inflammation, leading to tissue necrosis with the concomitant damage of bone trabeculae and the bone matrix. Counterbalancing the bone loss occurring in the infected areas, periosteal apposition and new bone formation occur [51].

Although each type of bone infection is established differently, the most frequent causative agent is *Staphylococcus aureus* [9]. Coagulase-negative staphylococci, Gram negative bacilli, mycobacteria, and streptococci are other pathogens involved in osteomyelitis [51].

For a pathogen to colonize the bone, it has to be able to adhere, penetrate, and survive within bone cells. Staphylococci have been shown to use receptors to bind the host fibronectin and integrins, which are expressed either on the bone matrix or by bone forming cells [55,56]. *S. aureus* has been shown to colonize and persist not only on osteoblasts [57,58] but also in their progenitors, including MSC, and in their downstream mature cell, the osteocytes [59,60,61,62].

The persistence of the bacteria inside bone-forming lineage cells leads to their cell death [63,64], which is caused by virulence factors such as alpha-type phenol/soluble modulins (PSMs) [56,65]. In order to persist within the hypoxic bone microenvironment [66], the adaptation of *S. aureus* is regulated by the staphylococcal respiratory response (SrrAB) that coordinates the bacterial survival under hypoxia and also virulence and cytotoxicity towards host cells [67]. However, the chronic persistence of *S. aureus* in the bone is regulated by the staphylococcal global stress regulator SigB, which controls the amount of virulence factors produced in order to escape the immune surveillance, avoiding the formation of abscesses that conduct to the pathogen elimination [68,69]. However, osteoblasts are not inert niches for the invading bacteria, as in vitro studies have pointed to the recognition of unmethylated CpG-DNA through toll like receptor (TLR) 9 and subsequent reactive oxygen species (ROS) production, leading to the killing of intracellular bacteria by oxidative stress [70]. Oxidative stress has a dual role on bone metabolism during infections. On one hand, the production of ROS such as O_2_^−^, H_2_O_2_ by NADPH oxidase and myeloperoxidase may directly kill the invading pathogen or facilitate the pathogen elimination by other mechanisms [71]. On the other hand, the production of ROS has deleterious effects on bone integrity, causing osteolysis by increasing osteoclast function and decreasing osteoblast differentiation [72]. The production of ROS such as NO and O_2_^−^ upon RANKL stimulation is important for the normal osteoclast differentiation. Moreover, pro-inflammatory cytokines such as Tumor Necrosis Factor alpha (TNFα) and interleukin 1 beta (IL1 β) enhance ROS production in osteoclast progenitors [73,74], which may also explain the increased osteoclastogenesis observed during inflammatory conditions. In osteoblasts, while basal levels of ROS are required for osteointegration, increased ROS production causes osteoblast apoptosis and decreased bone mass [72]. Therefore, the recognition of *S. aureus* by osteoblasts and subsequent ROS production may contribute to the immunopathology observed during osteomyelitis and, thus, is partly responsible for the observed osteopenia.

Besides colonizing and persisting in bone forming cells, staphylococci also interact with bone-degrading cells by invading them, which can have three possible outcomes: (1) intracellular bacterial proliferation [75], (2) osteoclast death [65], or (3) enhanced bone resorption [76]. The latter effect may also be potentiated by *S. aureus* protein A [77] and requires an intact host IL1R signaling pathway [78].

Bacterial osteomyelitis leads to bone loss not only through increased bone erosion but also by hampering bone formation. The decreased bone mass observed during osteomyelitis caused by *Borrelia burgdoferi*, the pathogen responsible for Lyme Disease, is due to a reduction in the number of osteoblasts in the trabecular bone without increased numbers of osteoclasts or bone degradation [7,79].

For other pathogens, the mechanisms leading to bone loss are not well stablished. Other well-known but rare causative agents of osteomyelitis are both tuberculous and non-tuberculous mycobacteria [8]. Vertebral osteomyelitis is a component of Pott’s disease, an extrapulmonary manifestation of Tuberculosis that often causes bone loss and neurological deficits [80,81]. Regarding non-tuberculous mycobacterial infection, *Mycobacterium avium* is described as the most frequent etiologic cause of vertebral osteomyelitis [82]. Tuberculous mycobacteria usually reach the bone through the hematogenous route, whereas non-tuberculous mycobacterial species have been described to colonize the bone either by hematogenous dissemination or by local and lymphatic spread from primary infectious sites [8]. The cellular hosts for mycobacteria within the bone environment are poorly described.

Overall, bone loss during osteomyelitis seems to be caused by the interaction of the invading pathogen with the bone cells, and the modulation of the host immune response, allowing the replication and persistence of the bacteria within an avascular environment, where antibiotics are not capable of penetrating.

## 4. Indirect Effects of Infection on Bone Metabolism

Microorganisms may disrupt bone homeostasis either directly, by infecting bone cells (described above) and/or indirectly through mediators produced by the pathogen and/or by the host. Interestingly, infection is not an absolute requirement for microorganisms to indirectly modulate the bone homeostasis. Commensals in the gut have been shown to induce bone loss by interacting with the immune cells, causing the release of osteoclastogenic cytokines [83]. Additionally, products of the pathogen metabolism may persist within the medullary cavity and elicit an inflammatory response that could disrupt bone homeostasis. For instance, hemozin, a byproduct of hemoglobin metabolism by *Plasmadium* causes a chronic inflammatory response that is sensed by osteoblasts, resulting in the increased secretion of RANKL and, hence, increased osteoclastogenesis and concomitant bone resorption [84].

The indirect mechanisms of bone metabolism disruption consist mostly of the pathogen’s interaction with the host immune system. However, immune-independent mechanisms also occur, namely, the alteration of vitamin D levels during infection. The impact of the disrupted immune homeostasis on bone metabolism has been well-documented in auto-immune disorders, namely, systemic osteoporosis, psoriatic arthritis, rheumatoid arthritis, and spondyloarthritis [85,86,87].

Usually, these pathologies are accompanied by an exacerbated production of immune modulators that activate osteoclasts, promoting bone resorption [11]. Along these lines, infection also involves the recognition of the invading pathogen and subsequent activation of the innate and adaptative immune system with the production of several inflammatory mediators such as cytokines and growth factors, which, in turn, might activate bone resorption and/or impair bone formation. Indeed, the levels of inflammatory mediators in the host are associated with the degree of bone loss [87]. As the production of immune mediators needs to be sustained in order to have significant effects on bone metabolism, it is expected that the sustained production of inflammatory cytokines during chronic infections might have a higher impact in the bone tissue. Bearing this in mind, the following section will cover potential modulators of bone metabolism during infection.

### 4.1. Immune Modulators

The immune response mounted to an ongoing infection may be divided into two components: the proinflammatory cytokines such as TNFα and IL1β; and anti-inflammatory cytokines such as IL11 [87,88,89]. These immune modulators are capable of altering bone homeostasis by acting on bone cells. Here, we will summarize the effects of these mediators in bone metabolism.

#### 4.1.1. TNFα

The pro-inflammatory cytokine TNFα is produced mostly by activated T cells and macrophages upon the recognition of the invading pathogen during an ongoing infection. This cytokine leads to an increase in the number of osteoclast precursors within the bone marrow and promotes their proliferation and differentiation, thus, contributing to an increased osteoclast activity [90].

On other hand, the effects of TNFα in osteoblastogenesis are dual, as it can induce or suppress osteoblast formation, depending on the stage of differentiation of the TNFα-sensing cell. At the earliest stages, MSC respond to TNFα with the differentiation towards the osteoblast lineage [91]. When the cytokine is sensed by cells at later differentiation stages, it has an anti-osteoblastogenic effect, inhibiting the osteoblast-specific transcription factor Runx*2*, thus impairing further differentiation into mature bone forming cells [92].

#### 4.1.2. IL1β

Another proinflammatory cytokine produced in response to an ongoing infection is IL1β. Besides its crucial effects in the immune response, it may also have a role in immunopathology with dual consequences in the bone homeostasis. IL1β signaling induces the upregulation of RANKL expression in osteoblasts, stimulating osteoclastogenesis and bone resorption. Simultaneously, the cytokine also causes increased expression of Wnt antagonists, thus hampering osteoblastogenesis [93,94,95].

#### 4.1.3. IFNγ

Similarly to IL1β, IFNγ is also an important mediator of the immune response to infection but when chronically elevated may have important immunopathological consequences. For instance, IFNγ disturbs hematopoiesis, resulting in the formation of short-lived hematopoietic cells, such as erythrocytes [96,97]. Therefore, an immunopathologic consequence of elevated levels of IFNγ is related with osteoclastogenesis. It has been described as directly anti-osteoclastogenic and indirectly pro-osteoclastogenic [98,99]. In case of infection and inflammation, IFNγ acts as a pro-osteoclastogenic agent leading to an increased bone resorption [100].

#### 4.1.4. IL6 and Related Cytokines

IL6 has also several effects on the body, controlling several processes in the response to infection. In bone metabolism, IL6 strengthens the effects of TNFα and IL1β, through RANKL-mediated signaling pathways, supporting the fusion of osteoclast precursors [101]. Another effect of IL6 is the increased release of endogenous glucocorticoids by the suprarenal glands upon activation of the hypothalamic-pituitary-adrenal axis [87]. Glucocorticoids may then induce bone loss by promoting osteoclastogenesis, via upregulation of RANKL and suppression of OPG production, thus enhancing bone resorption [102,103]. Simultaneously, glucocorticoids inhibit osteoblastogenesis by suppressing the Wnt signaling pathway, thus decreasing bone formation [104]. Furthermore, in vitro stimulation of bone-marrow-derived MSC with TNFα and IFNγ up-regulated the production of IL6 [30], suggesting that a local positive feedback mechanism could enhance osteoclastogenesis and subsequent osteolysis at the expense of new bone formation. Indeed, IL6 signaling in osteoblasts inhibits the formation of 5-alpha dihydrotestosterone (DHT), which is an anabolic hormone important for the formation of new bone tissue. Of note, the inhibition of bone formation by IL6 was reverted by doxycycline [105]. The importance of targeting IL6 levels and/or its signaling as a therapeutic approach gains further importance when considering the evidence that this cytokine induces the osteoblast transition of vascular smooth muscle cells with the calcification of vessels and increased cardiovascular and renal risk [106]. The differential role of IL6 on osteoblast formation and/or differentiation seems to depend on the type of cell receiving the signal, which needs to be further explored.

IL6 belongs to a large family of cytokines that includes IL11. The latter regulates positively and in a direct manner the differentiation of osteoblast from mesenchymal progenitors. IL11 has been demonstrated to activate Wnt signalling, inducing the differentiation of osteoblast progenitors into osteoblasts [107].

### 4.2. Indirect Modulation of Immune Response

#### 4.2.1. Vitamin D

As mentioned before, vitamin D is a major player in calcium metabolism and consequently in bone mineralization. Another increasingly recognized role of vitamin D is the regulation of both innate and adaptative immunity [108]. Even though vitamin D itself has an impact on bone homeostasis, here, we will focus on the role of vitamin D during infection as an immune modulator and its consequences for bone homeostasis. Vitamin D is hydroxylated in the body, by the enzyme CYP27B1, forming its active form, calcitriol [109]. CYP27B1 expression has been shown to be upregulated during infection by *Mycobacterium tuberculosis*, the causative agent of tuberculosis, leading to the increased synthesis of calcitriol [108,110]. Calcitriol also promotes the production of the anti-inflammatory cytokines IL4, IL5, and IL10, while decreasing the production of pro-inflammatory cytokines such as IFNγ, IL6, TNFα [111,112]. Moreover, vitamin D is capable of hampering IL1β, thus contributing to tissue preservation, due to the reduction in the immunopathology induced by IL1β [113]. In bone cells, CYP27B1 expression is not regulated by PTH signaling, but by high calcium concentrations. In this way, increased synthesis of vitamin D in the bone enhances bone matrix mineralization [114]. Furthermore, there is evidence that increased circulating levels of vitamin D amplifies the pro-osteoblastogenic effects of insulin growth factor-1 (IGF1) in MSC, promoting osteoblast differentiation [115]. On the other side, vitamin D impairs osteoclast differentiation by decreasing the number of its precursors in bone marrow [116].

#### 4.2.2. PTH

PTH is an important regulator of bone homeostasis, by increasing osteoblast-mediated bone formation. Consequently, its use to revert the decreased osteoblast number and activity during sepsis may be a potential therapeutic intervention to bone loss. Sepsis is a life-threatening condition in which the immune response to an ongoing infection causes a pro-inflammatory storm that is followed by immune suppression. Recently, using a murine model of sepsis, it was found that the osteoblast number and activity were suppressed during sepsis but the administration of PTH rescued bone development [10].

#### 4.2.3. Growth and Differentiation Factors

Transforming growth factor-beta (TGF-β) and bone morphogenetic protein-2 (BMP-2) are important for bone formation during homeostasis [117]. However, their effect in bone metabolism is overcome by IL1β production upon stimulation with lipopolysaccharide (LPS), a component of the bacterial wall [118].

M-CSF is an important inducer of osteoclastogenesis and it is also produced during infection in order to replenish the myeloid cells spent in the fight against the invading pathogen [119]. Indeed, during a chronic inflammatory status such as ankylosing spondylitis and rheumatoid arthritis, M-CSF is produced by stromal cells in the bone marrow in response to TNFα, increasing the numbers of osteoclasts and subsequent osteolysis [120,121,122,123]. Therefore, it is plausible that during infection, direct and indirect mechanisms lead to elevated systemic and/or local level of M-CSF, impacting osteoclastogenesis, with deleterious consequences for bone homeostasis.

Besides TGF-β and M-CSF, other growth factors such as growth differentiation factor 8 (GDF8) have a role in bone metabolism [124]. Further studies are required to understand whether infection affects the production and/or levels of these molecules and the concomitant impact on bone metabolism during infection.

In summary, immune mediators produced during chronic infection may interfere with bone metabolism, mostly creating an imbalance between bone formation and degradation, resulting in bone loss (Figure 4). The effect may be counteracted by endogenous mediators such as vitamin D and PTH which may have useful applications in therapeutic protocols.

## 5. Conclusions

Bone homeostasis is tightly regulated, and microorganisms such as gut commensals or pathogens have important interactions with bone metabolism. The actions of pathogens in bone metabolism may be both direct and indirect. The direct mechanisms involve the infection of bone-related cells, leading to their death, whereas the indirect mechanisms may involve the production of pro-inflammatory cytokines such as TNFα, IL1β, IL6 and the growth factor M-CSF, which in turn may increase osteoclast formation and activity while impairing the formation of new bone tissue. Endogenous mediators such as vitamin D and PTH counterbalance bone loss, protecting and restoring the bone mass. Further studies are required to identify other mediators that can be protective and restore bone mass. Overall, both direct and indirect mechanisms cause bone loss.

The decreased bone mass due to chronic infection is a relevant aspect of immunopathology and has important consequences to the patient’s quality of life. Therefore, the identification of the molecular and cellular interactions between pathogens and bone cells will lay the groundwork for further investigations and new tools to diagnose and treat bone loss induced by infection, as these are of critical importance in the improvement of the clinical management of these pathologies.

## Figures and Tables

**Figure 1 microorganisms-08-01765-f001:**
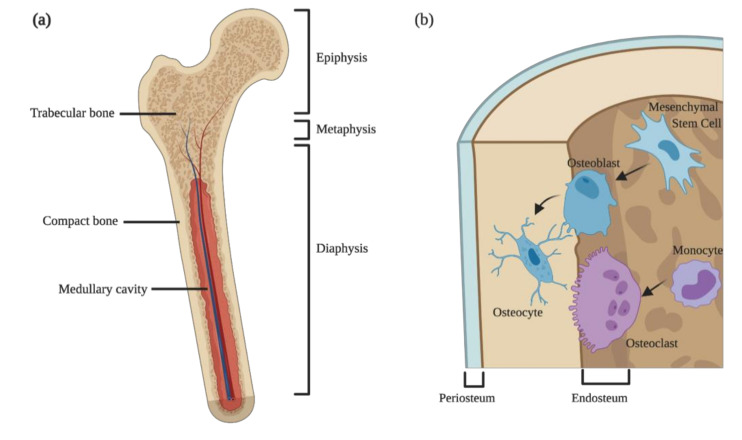
Bone structure and cellular components. (**a**) Anatomical division and types of bone present in long bones; (**b**) cellular components of bone and their cellular precursors. Osteoblasts are derived from mesenchymal stem cells and form new bones by the deposition of extracellular and mineral matrix. When osteoblasts become entrapped in the newly formed bone, they mature into osteocytes. Bone degradation is accomplished by large multinucleated cells, the osteoclasts, which have a myeloid origin.

**Figure 2 microorganisms-08-01765-f002:**
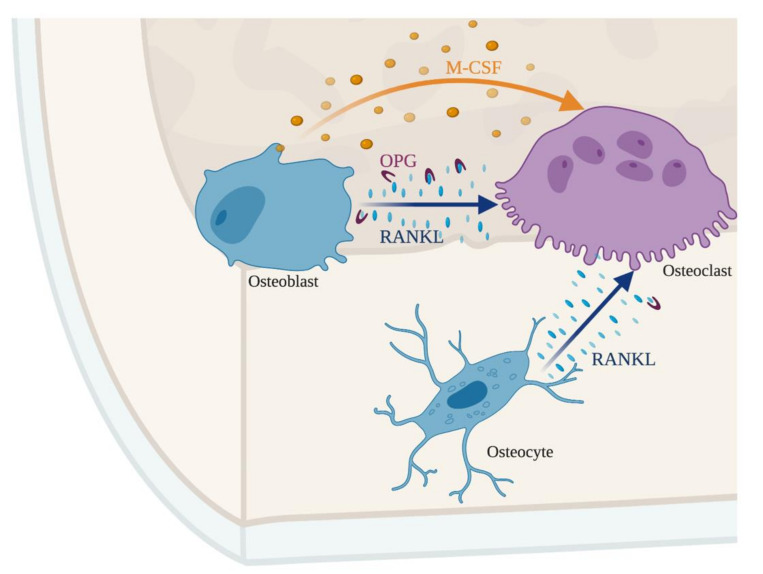
The crosstalk between osteoblasts, osteocytes, and osteoclasts maintains an adequate bone mass. The balance between receptor activator of nuclear factor kappa-B ligand (RANKL) and macrophage-colony stimulating factor (M-CSF) controls osteoclastogenesis. Osteoblast and osteocytes produce RANKL that binds to its receptor on the surface of osteoclasts. Simultaneously, osteoblasts secrete osteoprotegerin (OPG), a decoy receptor for RANKL. The production of OPG makes RANKL unavailable for osteoclasts, representing a second layer controlling bone resorption, dependent on osteoblasts.

**Figure 3 microorganisms-08-01765-f003:**
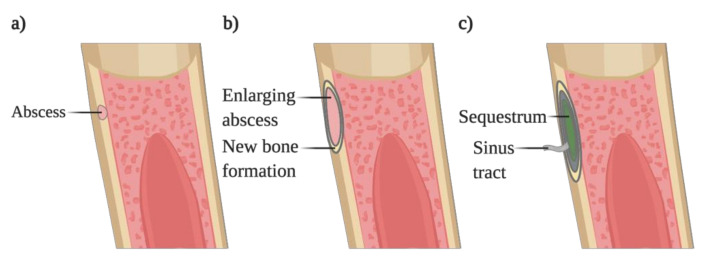
Progression of bacterial osteomyelitis. (**a**) Pathogens colonize the cortical bone, forming an abscess. (**b**) The forming abscess increases in size and is surrounded by new bone formation, causing the elevation of the periosteum. (**c**) The localized infection constricts the vascularization leading to the necrosis of the bone, forming the sequestrum. The sequestrum is surrounded by new bone where a sinus tract will be formed, allowing the drainage of the pus.

**Figure 4 microorganisms-08-01765-f004:**
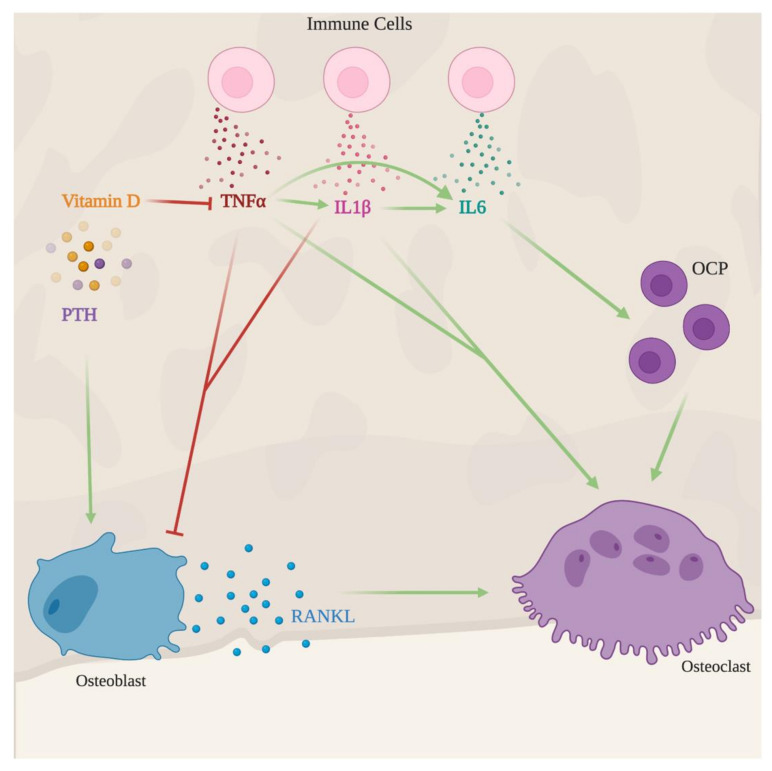
Indirect effects of infection in the bone. Upon recognition of an invading pathogen, immune cells produce several immune mediators as IL1β, TNFα, and IL6. These three cytokines promote osteoclast maturation and activity, hence favoring bone resorption. IL1β and TNFα further promote bone loss by inhibiting new bone formation. Vitamin D and parathyroid hormone (PTH) counterbalance infection-induced bone loss. The former mediator decreases the production of pro-inflammatory cytokines such as TNFα, IL1β and the latter hormone rescues osteoblast development and, thus, leads to new bone formation.

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
