# Peer review of "The Crossroads between Infection and Bone Loss"

_microorganisms, 2020, doi:10.3390/microorganisms8111765_

Round 1
Reviewer 1 Report
In this interesting review, the authors discuss the actions of pathogens in bone metabolism and the underlying mechanisms. First, they address the impact of infection in bone metabolism to understand the pathways of bone formation degradation. In further parts, they discuss the mechanisms regulating these pathways. Finally, the mechanisms responsible for bone loss during infection are discussed.
This is a very well written review, which gives us a bright overview over the relationship between infection and bone loss underlying mechanisms. I have only one minor remark, which I believe is important for the completion of this review. The mechanism of reactive oxygen species (ROS) production, leading to the killing of intracellular bacteria by oxidative stress is an important protection mechanism for osteoblasts and other cells, which are involved in ossification. This part should be extended and furthermore the molecular mechanism underlying ossification and inflammation could be elucidated upon further as shown in some publications in detail (PMID: 25159306; PMID: 29228352; and PMID: 31781089).
Author Response
Reviewer #1
In this interesting review, the authors discuss the actions of pathogens in bone metabolism and the underlying mechanisms. First, they address the impact of infection in bone metabolism to understand the pathways of bone formation degradation. In further parts, they discuss the mechanisms regulating these pathways. Finally, the mechanisms responsible for bone loss during infection are discussed.
This is a very well written review, which gives us a bright overview over the relationship between infection and bone loss underlying mechanisms. I have only one minor remark, which I believe is important for the completion of this review. The mechanism of reactive oxygen species (ROS) production, leading to the killing of intracellular bacteria by oxidative stress is an important protection mechanism for osteoblasts and other cells, which are involved in ossification. This part should be extended and furthermore the molecular mechanism underlying ossification and inflammation could be elucidated upon further as shown in some publications in detail (PMID: 25159306; PMID: 29228352; and PMID: 31781089).
We thank the reviewer for his/her comments and suggestions. In order to address the remark about ROS production and ossification, we expanded the role of ROS production in the bone (lines 275 to 288). Regarding the mechanisms of inflammation and ossification we have extended the topic on the manuscript. For structural reasons, we decided to not create an independent section on inflammation and ossification and discuss how inflammation impairs ossification throughout the text (lines 122 to 125 and lines 384 to 394).
Reviewer 2 Report
As text, this is a generally well-written work, easy to comprehend and follow. References are well selcted. However, as a review work, it should be addressed to a specific addressee (or group of recipients), however, after reading this work, I am not able to conclusively say to whom it is addressed, therefore in is not easy to judge their validity and usefulness for a reader interested in that specific sector. There is no information on how to treat the infection, or to what extent the changes caused by the infection are reversible. The introduction must be expanded a little to clarify what the aims of this review are, present a clearer, better focused hypothesis. Finally, the conclusions are a little too reductive with respect to what has been described in the body of the manuscript.
The second major the major drawback of the present manuscript is the language. Especially the review should be characterized with a correct language, however, in this manuscript you can often find not entirely correct linguistic expressions. Just to some form the first sections:
L61 “elevated number of cells” (in relation to what?)
L62 “changing mechanical demands” (mean “loads”?)
L66 frontier?
L70 “regions of insertion” (mean “attachment”?)
Also some names used are abbreviated , while some not, especially in the second part of the manuscript (e.g. Fas ligand = FasL; fibroblast growth factors = FGF; alpha-type phenol-soluble modulins = PSMs or PSMalpha ).
Please proof-read the whole manuscript carefully one again.
Author Response
Reviewer #2
As text, this is a generally well-written work, easy to comprehend and follow. References are well selcted. However, as a review work, it should be addressed to a specific addressee (or group of recipients), however, after reading this work, I am not able to conclusively say to whom it is addressed, therefore in is not easy to judge their validity and usefulness for a reader interested in that specific sector.
We thank the reviewer for his/her comments. We have altered the introduction in order to clearly address potential readers of the manuscript (lines 54-60), which we consider to be basic research scientists working on immunology or microbiology.
There is no information on how to treat the infection, or to what extent the changes caused by the infection are reversible.
Regarding the lack of information on the clinical management of bone infections and its consequences, we consider that they are out of the scope of this work and should be reserved for clinical reviews.
The introduction must be expanded a little to clarify what the aims of this review are, present a clearer, better focused hypothesis. Finally, the conclusions are a little too reductive with respect to what has been described in the body of the manuscript.
As requested, we extended the introduction to better identify the aims of the manuscript (lines 38-39 and 54-60). The conclusion was also modified in order to include all content covered throughout the manuscript (lines 457 to 469).
The second major the major drawback of the present manuscript is the language. Especially the review should be characterized with a correct language, however, in this manuscript you can often find not entirely correct linguistic expressions. Just to some form the first sections:
L61 “elevated number of cells” (in relation to what?)
We rewrote to “high numbers of cells compared to other tissues in the body” (line 69)
L62 “changing mechanical demands” (mean “loads”?)
We altered to “loads” (line 70)
L66 frontier?
Replaced to boundary (line 75)
L70 “regions of insertion” (mean “attachment”?)
Changed to “attachments” (line 79)
Also some names used are abbreviated , while some not, especially in the second part of the manuscript (e.g. Fas ligand = FasL; fibroblast growth factors = FGF; alpha-type phenol-soluble modulins = PSMs or PSMalpha ).
We have abbreviated the aforementioned names lines and several others (lines 211, 218, 265, 273, 274).
Please proof-read the whole manuscript carefully one again.
We believe the reviewer meant “once again”. We thank the reviewer for the advice and have proof-read the whole manuscript.
Round 2
Reviewer 2 Report
I would like to thank the Authors for considering the comments mentioned in the previous revision. Now, the opening paragraphs are clearer. Valuable addition, thank you. Although they assured that they read the entire manuscript again, it's hard to believe that they didn't notice any other language errors or typos in it. Some examples ?
L72 Should be “macroscopic and the microscopic levels”.
L95 “zooming in in”
L213 “pos-menopausal”
Once again, I would recommend check the English language and the text throughout the manuscript. This is not any “mean” suggestion, the absence of linguistic errors really improves the quality of publication.
Author Response
I would like to thank the Authors for considering the comments mentioned in the previous revision. Now, the opening paragraphs are clearer. Valuable addition, thank you. Although they assured that they read the entire manuscript again, it's hard to believe that they didn't notice any other language errors or typos in it. Some examples?
We appreciate that the reviewer acknowledged the improvements made on the manuscript. We missed some typos as in “post-menopausal” and we apologize for it.
L72 Should be “macroscopic and the microscopic levels”.
We changed “configurations” for “levels” as requested by the reviewer.
L95 “zooming in in”
We altered to “Zooming in on the microstructure”.
L213 “pos-menopausal”
We added the missing t.
Once again, I would recommend check the English language and the text throughout the manuscript. This is not any “mean” suggestion, the absence of linguistic errors really improves the quality of publication.
Thank you for your recommendation. We have checked the English language and the text throughout the manuscript.